# Applying Digital Twin and Multi-Adaptive Genetic Algorithms in Human–Robot Cooperative Assembly Optimization

**Doan Thanh Xuan, Tran Van Huynh, Nguyen Thanh Hung** and **Vu Toan Thang** *

School of Mechanical Engineering, Hanoi University of Science and Technology, Hanoi 100000, Vietnam
* Correspondence: thang.vutoan@hust.edu.vn; Tel.: +84-912051155

**Abstract:** In this study, we utilized digital twin technology in combination with genetic algorithms to optimize human–robot cooperation in a miniature light bulb assembly production line. First, the digital twin was used to find the robot's motion trajectory; a digital replica of the assembly system and human was created by combining sensors that track the position and activity characteristics of the human in the workspace, which helped to prevent human–robot conflicts. Then, a multi-adaptive genetic algorithm was applied to calculate optimal ergonomics and create a worker's movement schedule. To ensure continuous operation and no shortage of materials, the worker must observe and move to the input conveyor and material pallets to supply materials to the system. It aimed to provide more input materials for the assembly line while allowing the worker's task to take place in parallel with the robotic assembly operation. The algorithm was designed to reduce the number of moves required to obtain materials and to ensure that the robot always had enough materials to assemble along the defined trajectory, thus, saving labor and optimizing the manufacturing process. The combination of a digital twin and multi-adaptive genetic algorithm optimized the robot's movement path and the number of movements performed by the human operator in parallel.

**Keywords:** multi-objective optimization algorithm; human–robot cooperation; production process

## 1. Introduction

A manufacturing system is capable of performing various tasks and utilizing different types of resources. In order to adapt to the modern manufacturing landscape, the concept of human–robot cooperation systems has gained increasing popularity due to the potential benefits of close human–robot interaction. Human–robot cooperation (HRC) combines the positive aspects of human and robot skills—human intelligence, creativity, flexibility, and ability to work with different tools in varying situations and robot speed, accuracy, durability, and high efficiency—to accomplish a shared task, such as an assembly, in a collaborative workspace [1].

The working environment of HRC is complex and dynamic, with the ability to expand and adapt to different configurations continuously during operation. As these changes are updated rapidly during the design, development, and operation phases, human safety is a crucial factor that needs to be considered while designing the HRC workspace [2,3]. In this paper, we present a simulation-based digital twin that can control a cooperative human–robot assembly. The digital twin is used to plan the trajectory and develop a program to control the robot.

Optimal ergonomics is a research area that many scientists are interested in exploring in relation to human activities. With the increasing value placed on human labor and the high costs associated with it, it is necessary to prioritize the health and regeneration of human labor power. This involves considering the ergonomic requirements for each step in a task and assigning appropriate physiological characteristics to humans or robots [4,5]. In a similar approach, scientists [6] studied the optimization of assembly processes, with

goals of cost and time savings. They took into consideration the characteristics of the robot during assembly.

The focus of this research is the assembly line, which is a common and complex problem that often requires solutions in factories. In the studies on building human–robot collaboration [7–12], the effectiveness of genetic algorithms was demonstrated. As a member of the evolutionary computations group, this algorithm is based on two well-known biological theories, heredity and evolution. By simulating genetic biology, this method provides approximate solutions to optimization problems that cannot be solved using conventional methods [13].

In this study, we optimize ergonomics by minimizing the number of human movements required between locations within the collaborative environment. We provide a method to create the most optimal movement plan for the worker to collaborate with the robot, which moves according to the programmed trajectory.

## 2. Experimental Layout Design

The experimental model used in this study is a miniature lamp assembly system which consists of a workstation with both human and robot manipulators. The assembly process is performed by a UR3 robot manipulator, which has six degrees of freedom, a payload capacity of 3 kg, and a reach of 500 mm. The workstation receives a sub-assembly from the previous station, performs the mounting of additional parts, and transfers the sub-assembly to the next station.

The UR3 robot is responsible for picking up sockets from the pallet and placing them in the punch hole, then picking up light bulbs and placing them in the same punch hole. The light bulbs are conveyed on a belt, while the sockets are stored in a pallet. After a stamping cylinder applies force to adhere the socket to the bulb, the UR3 robot picks up the finished product and places it on the conveyor belt to transport it to the warehouse. Figure 1A displays an actual image of the entire assembly system, while Figure 1B shows a schematic of the system.

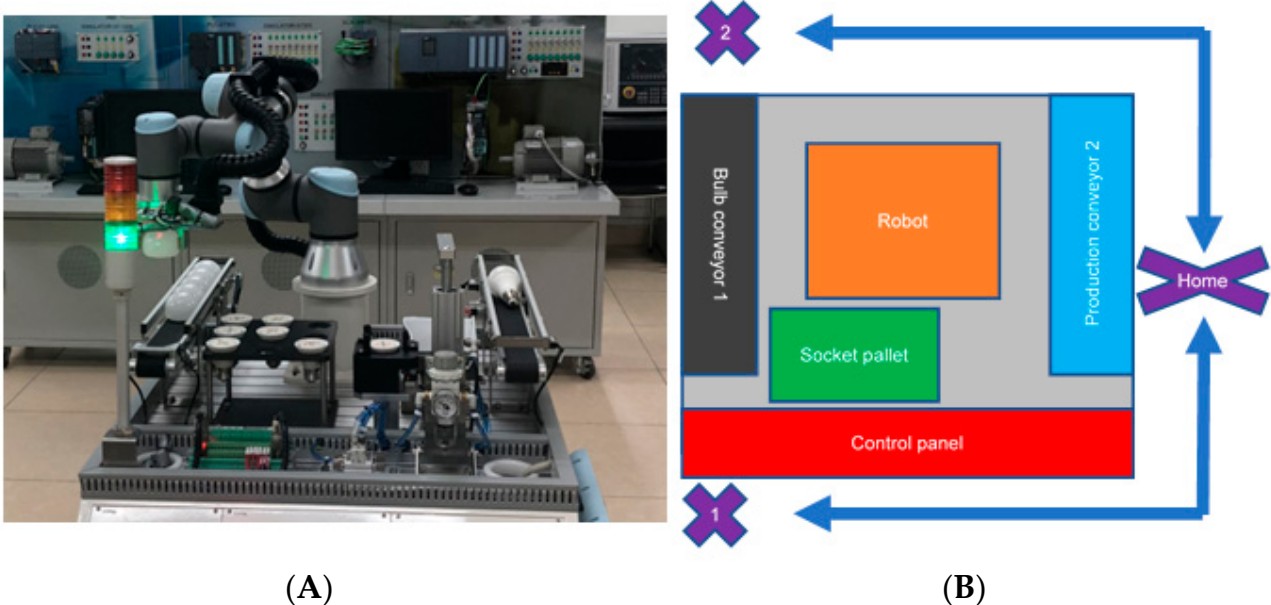

**(A)** **(B)**

**Figure 1.** Real picture (**A**) and diagram (**B**) of miniature lamp assembly system.

The worker in the assembly workstation performs three tasks: standing at the home position and bringing the finished product to the warehouse, placing the socket on the pallet when the number of socket materials is running low, and supplying more bulbs to the conveyor belt when the bulb materials are about to run out. Human and robot work in tandem to ensure that the system always has enough materials to operate smoothly

without interruption, and the finished product is delivered to the warehouse in a timely manner without causing congestion and stopping the system.

The sequence of steps is shown in Figure 2.

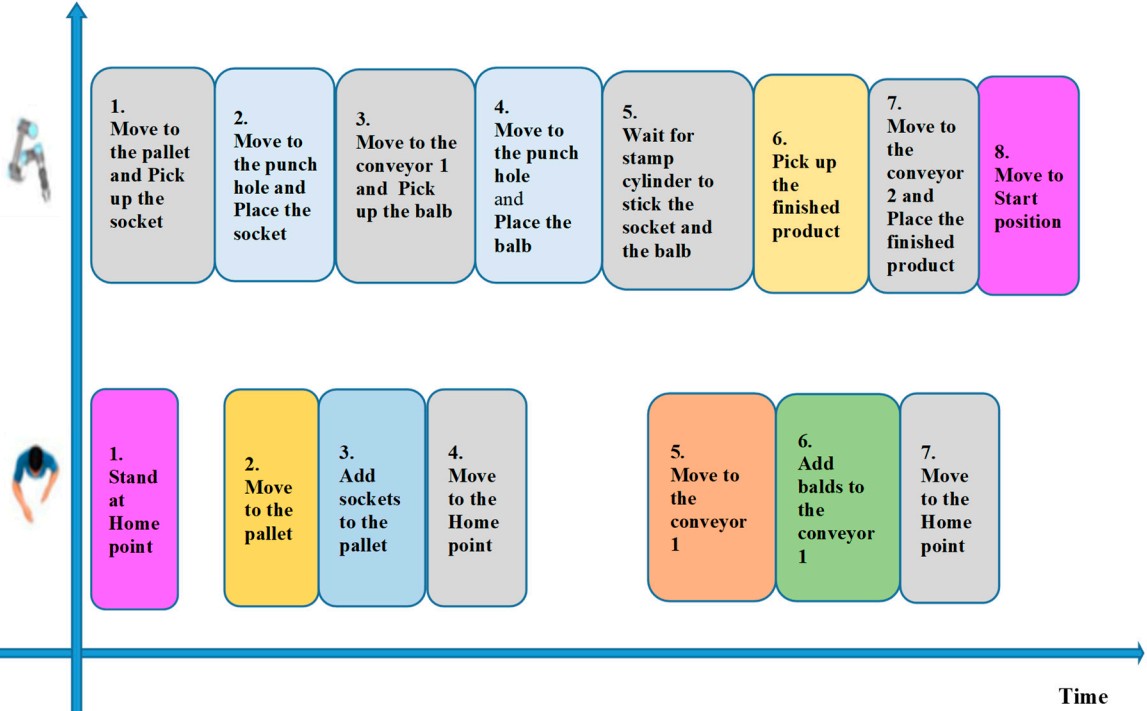

**Figure 2.** A sequence of tasks performed by humans and robots.

## 3. The Application of a Digital Twin to Find Robotic Motion Trajectory

The digital twin framework utilizes a dynamic simulation model which is created using Tecnomatix Process Simulate software and consists of three-dimensional computer-aided design (3D CAD) objects imported from a Siemens Support Center online library. The model is built in four layers—geometry, physics, behavior, and rule—as suggested by Tao [14]. Any additional components required at the workstation are created in NX as 3D objects and imported into the Tecnomatix environment. The Process Simulate Human package is used to create a digital model of the human worker for ergonomic evaluations, using realistic male and female human figures with deformable mesh technology to accurately represent body shapes [15]. The selected human model has a height of 160 cm, BMI of less than 25, and waist-to-hip ratio (for females).

This study explores the use of a Kinect sensor to monitor human positions and presence in a workspace, with the goal of using decision trees (DT) in human–robot cooperation (HRC) to track and analyze interference volumes between humans and robots, as well as the frequency of interferences. This information is then used to periodically optimize robot trajectories in order to account for areas where humans frequently enter. By using historical data of human positions, the simulation is able to self-learn and generate robot trajectories that are free from possible human intervention.

Figure 3 illustrates the images of the real human and robot and their digital counterparts in the digital twin. The system's digital model was built in Tecnomatix software. The red rectangle frame in the image indicates the positions where collision may occur between the human and robot while sharing the workspace.

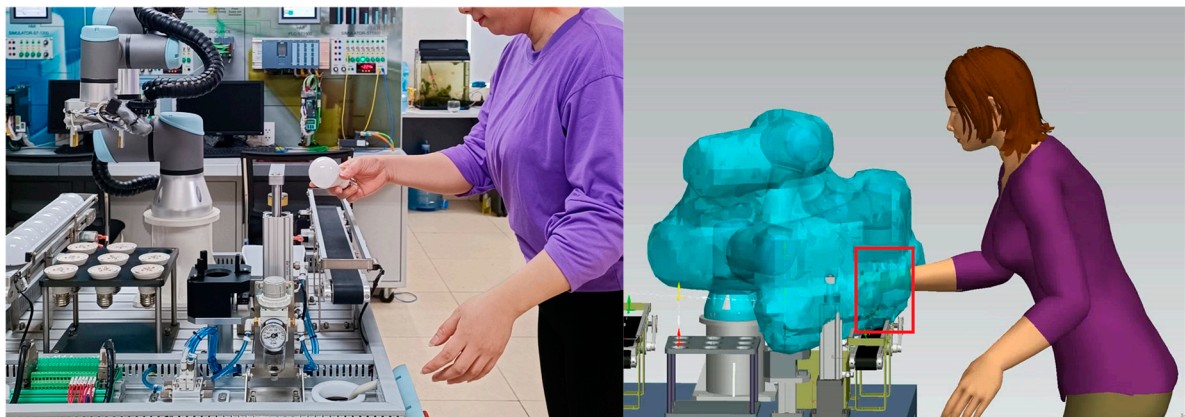

**Figure 3.** Real human and robot image (**on the left**)—human and robot in Tecnomatix software (**on the right**).

The DT method involves saving key positions for the robot in the simulation model. Once the tasks are identified and robotic tasks are assigned, they are simulated in a virtual environment to generate robot trajectories. Intermediate positions are generated automatically, and the robot avoids possible collisions with equipment. These paths are then saved in an SRC file, which can be loaded to the robot. In cases of dynamic changes, robot instructions are sent in the form of a message that overrides the original robot program [16].

## 4. Application of a Multi-Adaptive Genetic Algorithm to the Problem of Human–Robot Cooperation

### 4.1. Description of the Problem and Initial Conditions

The problem for the miniature light bulb assembly system is minimizing the number of times the worker moves, as long as the robot's operation is not interrupted by the lack of a bulb socket or bulb.

At the initial time ($t = 0$), it is assumed that the worker is standing at the Home position. On the pallet, there are enough S sockets, and on the conveyor belt, there are B bulbs.

The total time taken by a worker to provide sockets is as follows:

$$T_s = \sum_{i=1}^{s} Ts_i + 2T_m \tag{1}$$

$T_s$ represents the total time it takes for the worker to complete a socket supply. $T_m$ is the time it takes for the worker to move from the Home position to Position 1 and then return to the Home position.

$Ts_i$ is the time it takes to supply the $i$th socket, where $S$ represents the total number of sockets supplied in a single movement. Similarly, we can obtain the time it takes to supply bulbs to the assembly line using the following equation:

$$T_b = \sum_{j=1}^{b} Tb_j + 2T_m \tag{2}$$

$T_b$ represents the total time it takes for the worker to provide bulbs. $T_m$ is the time it takes for the worker to move from the Home position to Position 2 and then return to the Home position. The time interval for this movement is approximately the same as the time it takes the worker to move from the Home position to Position 1 using a defined motion trajectory.

By observing the actual activity of the worker and the robot through a camera, we can determine that the operating speed and acceleration of the robot are 300 mm/s and 1300 mm/s$^2$, respectively, and the average moving speed of the worker is 0.54 m/s. It is worth noting that the average time to complete the assembly of a single light bulb is 30 s.

*4.2. Designing a Multi-Adaptive Genetic Algorithm*

A schematic diagram of the multi-adaptive genetic algorithm (MGA) is shown in Figure 4 [17].

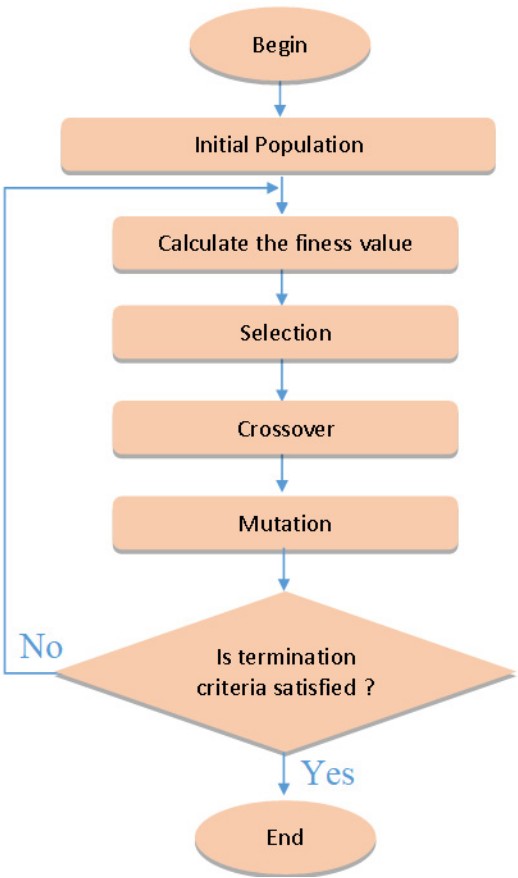

**Figure 4.** Genetic algorithm diagram.

A genetic algorithm applies the process of evolution in nature to solve optimization problems in practical applications. Starting from an initial population, an optimization function is created, and the optimal value is calculated to select the best-performing individuals. Then, the process of crossover and the impact of the mutation operator are performed to generate a new set of individuals with better optimal values. The process is repeated many times to allow for evolution. The final solution is an approximate optimal solution.

4.2.1. Chromosome Structure

'N' is the number of light bulbs that need to be assembled. The number of genes in a chromosome is twice the number of bulbs to be assembled, which is $2n$. Each gene corresponds to a time slot of 30 s, which is the completion time of assembling a single light bulb. For instance, if we need to assemble 100 light bulbs, the chromosome consists of 200 genes, each of which corresponds to a 30 s time slot.

The first half of the chromosome indicates whether the worker is stationary or moving. The value '0' means that the worker is stationary at the Home position, while the value '1' means that the worker is moving to either Position 1 (socket providing) or Position 2 (bulb providing). The second half of the chromosome represents the direction of the worker's movement. The value '1' corresponds to the worker moving to Position 1, and the value '0' corresponds to the worker moving to Position 2.

The first gene in the chromosome decides whether the worker moves from the Home position or not, and the $(1 + n)$th gene determines the direction of movement at the beginning of the first 30 s. Therefore, the pair of genes $(i)$th and $(i + n)$th determines

whether the worker moves and which direction to move at the beginning of the $(30 \times i)$th seconds. Chromosomes are paired as shown in Figure 5.

| 1 | 0 | 1 | ... | 1 | 1 | 1 | 0 | ... | 0 |
|---|---|---|-----|---|---|---|---|-----|---|

Gen position  1  2  3  ...  100  101  102  103  ...  200

**Figure 5.** A random chromosome.

The evaluation of the optimization criteria is based on the following parameters:
Es: this parameter assesses the level of available sockets on the pallet;
Eb: this parameter evaluates the level of available bulbs on the conveyor belt;
M: this parameter represents the optimal rating for the number of worker movements.
The formula for determining the Es index is:

$$\text{Es} = \sum_{k=1}^{n} s_k \tag{3}$$

For each gene index $k$ ($1 \leq k \leq n$) on the chromosome, the value $s_k$ is calculated as follows:

If the left gene at position $k$ has a value of 0 (the worker is not moving), or both left genes at positions $k$ have a value of 1 and the corresponding right gene at position $k + n$ has a value of 0 (the worker is moving and goes to Position 2 to provide bulbs), then $s_k$ is set to 0.

If the left gene at position $k$ and the corresponding right gene at position $k + n$ both have a value of 1 (the worker moves to provide sockets), then $s_k$ is calculated.

$$s_k = \begin{cases} r_s, & r_s \leq S \\ 1/r_s & > S \end{cases} \tag{4}$$

The time interval between the worker's movement to Position 1 (to provide sockets) at the $(k - 1)$th and $k$th times is denoted by $r_s$. The condition for avoiding a lack of sockets on the pallet is that the interval time between these two moves $r_s$ cannot be less than the time interval it takes the robot to assemble a number of bulbs equal to the maximum number of sockets on the pallet $S$. For instance, if the pallet can hold up to nine sockets, then $S = 9$. Therefore, $r_s$ is equal to the distance between the two genes responsible for moving to the pallet. If there are more than nine genes responsible for moving to the pallet, the robot is considered to be in a situation of material shortage and stops working. In this case, the $s_k$ parameter decreases and receives the inverse value of the $r_s$ parameter.

The index for assessing whether there are enough bulbs on the conveyor belt (Eb) has the same role as Es, and the formula for determining it is similar. The formula for determining the Eb parameter is:

$$\text{Eb} = \sum_{k=1}^{n} b_k \tag{5}$$

where:

$$b_k = \begin{cases} r_b, & r_b \leq B \\ 1/r_b, & r_b > B \end{cases} \tag{6}$$

The time it takes for the worker to move between the $(k - 1)$th and $k$th positions for providing bulbs is denoted as $r_b$. The maximum number of bulbs on the conveyor belt is $B$, and the value of $r_b$ is determined in a similar manner to $r_s$.

The total number of stationary human gene positions M is calculated using the formula:

$$\text{M} = 100 - \sum_{1}^{n} x_m \quad (\text{with } x_m = 1) \tag{7}$$

The genes encoded in the first half of the chromosome with a value of 1 correspond to the worker who moves during the assembly process. With a total time slot of 100, the number of genes assigned to the stationary worker is obtained by subtracting the number of genes assigned to the moving worker from 100. A greater number of stationary worker

genes can be achieved by reducing the number of moving worker genes. The goal is to maximize the value of M. Pareto efficiency refers to the optimal allocation of resources in order to achieve the most efficient outcome [18]. The fitness function, which encompasses all objectives, is formulated as follows:

$$f(x) = \sum \theta_x \, \omega_x f_x \tag{8}$$

In the multi-adaptive optimization problem, $\omega\alpha$ represents the weight of the $\alpha$th objective function, where the sum of all weights is equal to 1. The coefficient $\theta\alpha$ is used to adjust the range of values for the $\alpha$th objective function, which helps to obtain a balanced range of values for all objectives.

For this particular study, the weights for the three objectives are 0.3, 0.3, and 0.4, respectively. To ensure that all objectives have a similar range of variation, adjustment coefficients of 1, 1.5, and 1.28 are used for the first, second, and third objectives, respectively.

The formula assigns weights based on the priority of each problem to be solved. After conducting experiments on the model, the values in Formula (9) are selected. The two parameters Es and Eb have equal priority and are of lesser importance than M. The overall fitness function is calculated using the following formula:

$$f(x) = 0.3(\text{Es}) + 0.45(\text{Eb}) + 0.512(\text{M}) \tag{9}$$

To determine the optimal value of the chromosomes, a maximization approach is utilized. The chromosome with a higher optimal value of f(x) is considered to be better and is, thus, preserved across generations.

According to [19], the algorithm can be presented as below.

### 4.2.2. Initial Population

To begin the genetic algorithm, an initial population is randomly generated with randomly assigned genes. The resulting population has a certain level of diversity, but the optimality and fitness of the individuals are random. The size of the initial population greatly affects the efficiency of the algorithm. A small population with limited diversity may require many generations to produce optimal individuals, while a larger population with greater diversity may lead to faster convergence. However, it is important to balance the number of individuals with the number of genes and crossover rate to avoid duplications and save computational resources.

The initial population is created through a random process, but it is important to ensure that the initial population has sufficient diversity, meaning that the population entropy cannot be too small.

To represent and calculate population entropy, the solution space is divided into M non-overlapping regions ($Q_1$, $Q_2$, ... , $Q_M$). The probability that an individual in the population belongs to Qi is denoted by $p_i$ ($i = 1, 2, \dots , $ M). The population entropy of the $t$th generation ($S(t)$) can be expressed as:

$$S(t) = -\sum_{i=1}^{N} p_i \ln(p_i) \tag{10}$$

To estimate the population entropy, a range is used based on the fitness values instead of the solution space, where $F_{min}$ represents the minimum fitness value from the initial iteration to the $t$th generation, and $F_{max}$ represents the maximum value. An expansion coefficient $\alpha$ ($0 < \alpha < 0.1$) is used to expand the range (($1 - \alpha$)$F_{min}$, ($1 + \alpha$)$F_{max}$). The entire interval is then divided into $N$ regions, where $N$ is the number of individuals in the population. The number of individuals whose fitness value falls within the $i$th region is denoted by $l_i$ ($i = 1, 2, \dots , N$). The estimated value of $p_i$ is calculated based on these values.

$$\hat{p}_i = l_i / N \tag{11}$$

When the estimated value of $p_i$ is substituted into Equation (12), it provides an estimation of the population entropy $\hat{S}(t)$:

$$S(\hat{t}) = -\sum_{i=1}^{N} \hat{p}_i ln \hat{p}_i \tag{12}$$

### 4.2.3. Selection and Elitism

This study utilizes the roulette wheel selection [20] method to select individuals for the next generation, and the best individual in each generation is directly transferred to the next generation in the elitism step.

### 4.2.4. Crossover

The first half of the chromosome (left) indicates whether the worker moves or not at the start of the time slot, and the second half (right) determines the direction of the movement. Because there is a correlation between the two halves of the chromosome, it is necessary to ensure that the information structure is not broken during the crossover operation. After the first half of the chromosome is exchanged with the infectious chromosomes, the corresponding left half also needs to be exchanged with the corresponding chromosomes to maintain the information structure. This process is illustrated in Figure 6.

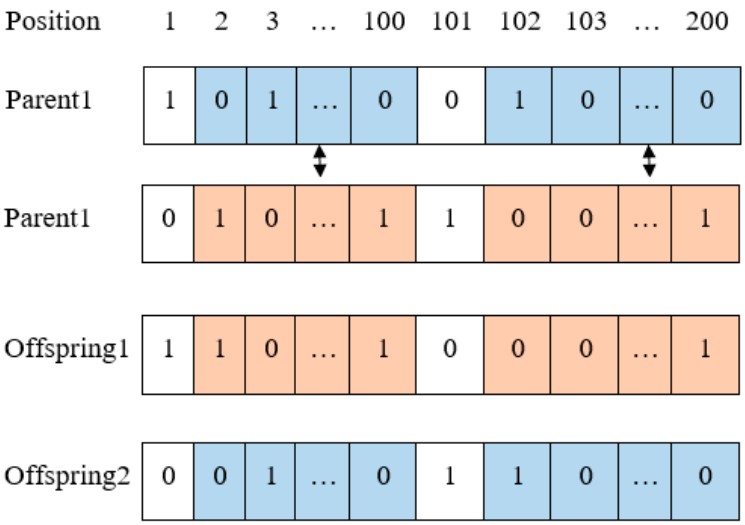

**Figure 6.** An example of crossover.

In this study, the crossover rate ($P_c$) is adaptively adjusted based on multiple factors. Initially, the basic crossover rate of the $t$th generation ($P_{c1}$) is determined using the population entropy, which is calculated by using the following equation:

$$\beta = S(t)/S_{max} \tag{13}$$

$$P_{c1} = P_{c2} + P_{c3}(1 - \beta) \tag{14}$$

The multi-adaptive adjustment of the crossover rate ($P_c$) in this study involves two steps. Firstly, the basic crossover rate ($P_{c1}$) of the $t$th generation is determined based on the population entropy, where $S_{max}$ is the maximum possible value of population entropy (i.e., $S_{max} = lnN$), and $P_{c2}$ and $P_{c3}$ are adjustable parameters. As population diversity decreases, the basic crossover rate increases to promote diversity.

Secondly, the crossover rate of an individual ($P_c$) is determined based on its fitness value using Equation (15):

$$P_c = P_{c1} F_{max}/(\gamma F) \tag{15}$$

The formula determines the crossover rate ($P_c$) based on the fitness value of the two individuals being crossed over. The larger fitness value, denoted by $F$, results in a lower crossover rate, preserving the structure of the better individual. To maintain diversity in the population, a coefficient $\gamma$ is used to increase the probability of individuals with lower fitness values entering the next generation.

### 4.2.5. Mutation

To preserve the diversity of both the population and individuals, a 0–1 variable ($X_{ij}^k$) was introduced. If individual $i$ and individual $j$ differ in the $k$th gene, then $X_{ij}^k$ equals 1; otherwise, $X_{ij}^k$ equals 0. The extent of diversity of the $k$th gene in all individuals of the $t$th generation population ($Y_t^k$) can be mathematically represented as:

$$Y_t^k = \sum_{i=1}^{N-1} \sum_{j=i+1}^{N} X_{ij}^k \qquad (16)$$

To determine the position for mutation in the mutation operation, the value of $Y_t^k$ is taken into consideration, where the probability of selecting the gene position for mutation increases as the genetic diversity decreases. Roulette wheel selection is used for this purpose. The mutation rate ($p_m$) in this study is also adaptively adjusted using a multi-step process. First, the basic mutation rate for the $t$th generation ($p_{m1}$) is calculated using 4.17, based on the population entropy.

$$p_{m1} = p_{m2} + p_{m3}(1 - \beta) \qquad (17)$$

The values of $p_{m2}$ and $p_{m3}$ can be adjusted as parameters, and, when the population diversity decreases, $p_{m1}$ is increased to promote the creation of new individuals and improve population diversity.

After this, the mutation rate of an individual ($p_m$) is determined based on the individual's fitness value using the following formula:

$$p_m = p_{m1}.F_{max}/(\gamma F) \qquad (18)$$

The mutation rate ($p_m$) of an individual is determined by its fitness value, as calculated by the following formula, where F is the fitness value of the individual after mutation. If the fitness value increases, the mutation rate is reduced to preserve the genes of high-performing individuals.

### 4.3. Results and Discussion

In Section 4.2, the results and discussion of an experiment involving the assembly of 100 light bulbs are presented. Initially, a worker performs the assembly without using a genetic algorithm and relied on the worker's observation and judgment to determine when to move to provide sockets and bulbs. Statistical data are collected over 30 rounds of observations, where 100 bulbs are assembled during each round, and the average number of moves made by the worker is found to be 42. Then, the same assembly task is performed using a genetic algorithm to determine when the worker moved and the direction of movement.

The equations to determine crossover rate and mutation rates are: $p_c = p_{c1}F_{max}/(\gamma F)$, $p_{c1} = p_{c2} + p_{c3}(1 - \beta)$, $p_m = p_{m1}F_{max}/(\gamma F)$, and $p_{m1} = p_{m2} + p_{m3}(1 - \beta)$.

During the actual experiment, appropriate parameter values are selected as follows: $p_{c2} = 0.6$, $p_{c3} = 0.3$, $p_{m2} = 0.04$, $p_{m3} = 0.06$, $\gamma = 2$, and $\alpha = 0.08$.

The MATLAB algorithm is executed 50 times with a population size of 100 for 500 iterations. The result of the average number of human movements is 36.48.

The performances of the algorithm in the experiment are shown in Figure 7.

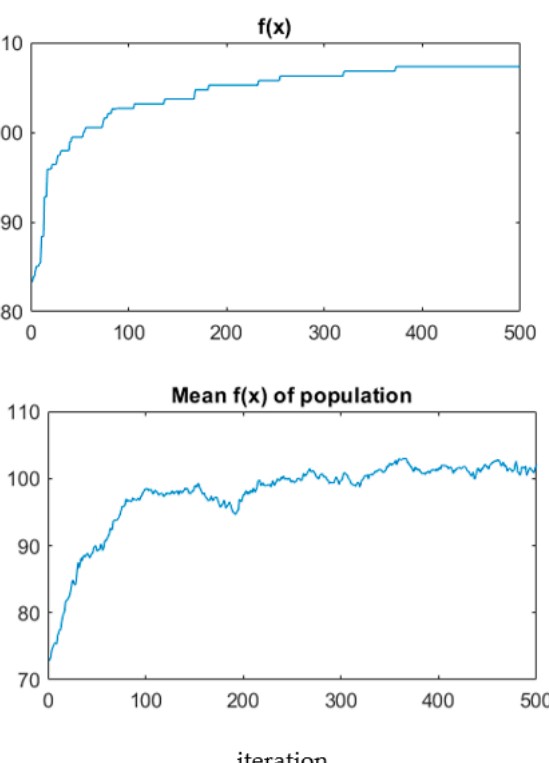

**Figure 7.** Performance of multi-adaptive genetic algorithm.

The graph indicates that the parameter f(x) consistently increases as generations progress, signifying that the individuals in the population improve over time.

The changes in population entropy are shown in Figure 8.

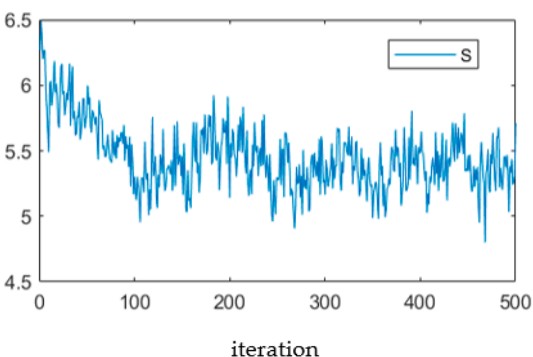

**Figure 8.** Generation-by-generation population entropy.

Figure 8 shows that the population's S value tends to decrease, although it fluctuates strongly. The fluctuation in entropy is mostly caused by mutations that make the population less stable. Mutations are necessary to increase the population's biodiversity. However, mutations also reduce the population's stability.

Entropy is a scientific concept and a measurable physical property used to indicate a state of disorder, randomness, or uncertainty. Entropy (S) reflects the stability and uniformity of a population.

We can observe that the entropy of the population is highly dynamic but tends to decrease over time. The fluctuations in entropy are mainly caused by the impact of mutations that disrupt the stability of the population. The presence of mutations is necessary as it increases the biological diversity of the population. However, excessive mutations can also lead to a reduction in population stability and imbalance in the population's various indices.

Figure 9 illustrates the variation of data for the mean and best individual values of the Es, Eb, M, and f(x) values.

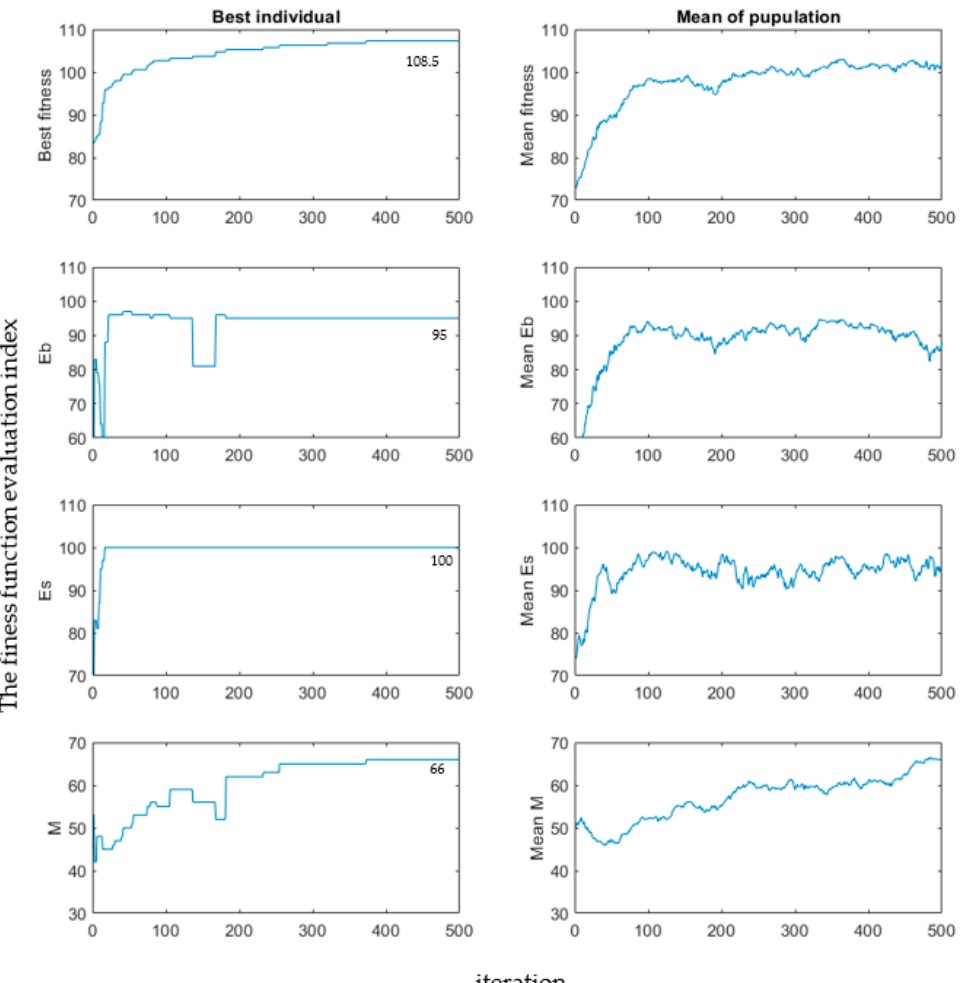

**Figure 9.** The miniaturized system's data changes.

The miniaturized system's data changes indicate the average and optimal values of the population for each genetic generation. The best individual for each generation shows strong fluctuations in the initial and competitive stages of the three optimal parameters. Once Eb (30%) and Es (30%) reach a steady state, the algorithm continues to adjust parameter M (40%) until it also reaches a steady state. As for the mean parameters, which are the population's average values for each genetic generation, they generally increase over time, exhibiting a similar trend to the best individual parameters, indicating that the population is moving towards the best individual.

The output is a graphical representation of the assembly process, which includes assigning tasks to the worker and collaborating with the robot to ensure efficient operation. The process is divided into 100 cells that represent 30 s time slots. The color of each cell indicates the corresponding task: white represents the worker standing still, green indicates going to the pallet to provide sockets, and red represents going to the conveyor belt to provide bulbs. Figure 10 illustrates this in detail.

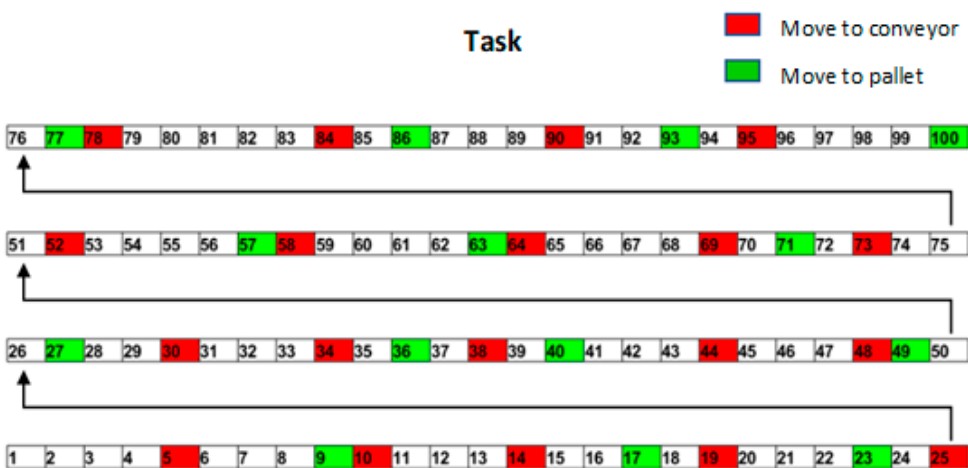

**Figure 10.** The optimized workflow of the best individual obtained from the algorithm.

Figure 10 illustrates the optimized workflow of the best individual obtained from the algorithm. The total working time is 3000 s. The process is divided into 100 cells that represent 30 s time slots. The color of each cell indicates the corresponding task; white represents the worker standing still, green indicates going to the pallet to provide sockets, and red represents going to the conveyor belt to provide bulbs. Figure 9 illustrates this in detail.

Cells 1, 2, 3, and 4, in white, represent the first 120 s (30 s × 4) during which the person stands still in the Home position and does not move. Cell 5, in red, represents the worker moving to the conveyor to supply additional bulb caps at second 121 and then returning to the Home position. Cells 6, 7, and 8, in white, represent the next 90 s (from second 151 to second 240), during which the worker stands still in the Home position. Cell 9 (from second 241 to 270), in green, represents the worker moving to the pallet to supply socket materials to the system. An integrated speaker system follows the signal of each time slot to notify the person whether to stand still or move and in which direction to move.

The worker activities are optimized to minimize movement. The time between movement activities is consistent and does not overlap to ensure that the robot has sufficient materials to operate.

Before using the multi-adaptive genetic algorithm, four people assemble 100 light bulbs in 50 rounds based on their observation ability, and a camera is used to record information. The average results of all four participants in the experiment are: the average number of times the cap is supplied is 25.85 times; the average number of times the tube is supplied is 21.03 times; the total number of movements is 46.87 times.

After 50 runs of the MATLAB software program using the MGA algorithm, the average number of cap supplies is 21.52. The number of tube supplies is 15.02, and the total number of movements is 36.54.

After applying the multi-adaptive genetic algorithm, the number of movements of the person decreases by about 22%.

## 5. Conclusions

In conclusion, this study successfully applied a digital twin and genetic algorithms to optimize the motion trajectory of a robot and improve human ergonomics in an assembly cell. The digital twin provided real-time tracking of human and robot activities to avoid collisions and determine the robot's optimal movement trajectory. The genetic algorithm was then used to minimize the total number of human movements and improve ergonomics by only requiring human movement when absolutely necessary. This was achieved while ensuring that there were always enough materials for continuous operation. The genetic algorithm was able to overcome the challenge of minimizing worker movement while also ensuring that the assembly system had enough materials, resulting in a significant reduction in the number of worker movements. Overall, the application of genetic algorithms in

human–robot collaboration proved to be effective in optimizing the number of human movements required in an assembly system.

The combination of a digital twin and genetic algorithms demonstrated in this study can be extended to optimize the collaboration between robots and humans in large-scale assembly. This approach enables better planning of robot layout and human resources, including the number and working time of workers, resulting in the optimization of the entire assembly line's operation.

In previous studies on the problem of human–robot collaboration in assembly tasks, the digital twin and multi-adaptive genetic algorithms were used separately and not in combination. The multi-adaptive genetic algorithm has not been applied or studied for computing human motion schedules that fit predetermined robot activity schedules. The weakness of our study is that it did not evaluate the impact of unforeseen errors that may occur in the system.

In the future, we will investigate the application of combining digital twins and multi-adaptive genetic algorithms on multiple robots operating simultaneously. Additionally, we will study the problem in the case of errors occurring outside the workflow of the system.

**Author Contributions:** D.T.X.: methodology, validation, formal analysis, data curation, writing—original draft preparation; T.V.H.: programming; V.T.T and N.T.H.: writing—review and editing, visualization, supervision. All authors have read and agreed to the published version of the manuscript.

**Funding:** This research received no external funding.

**Institutional Review Board Statement:** Not applicable.

**Informed Consent Statement:** Not applicable.

**Data Availability Statement:** The authors confirm that the data supporting the findings of this study are available within the article.

**Acknowledgments:** We would like to thank the Laboratory of Smart Digital Factory at the School of Mechanical Engineering—Hanoi University of Science and Technology—for their assistance in implementing this research.

**Conflicts of Interest:** The authors declare no conflict of interest.

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
