# Peer review of "Applying Digital Twin and Multi-Adaptive Genetic Algorithms in Human–Robot Cooperative Assembly Optimization"

_applsci, doi:10.3390/app13074229_

Round 1
Reviewer 1 Report
Authors mentioned that digital twin and multi-adaptive genetic algorithms in human-robot assembly work. However, it is hard to recognize the novelty in Abstract and conclusion sections. Previous research and motivation of the proposed research need to be emphasized. In addition, Figure quality looks unclear to be seen. In addition, authors must ask professional English service or ask native English colleagues. There are some comments as below.
1. Figure 2 font is so small.
2. Figure 3 quality is unclear to be seen.
3. In Figures 6,7, and 8, there are no units for x-axis.
4. In Figure 8, no unit for y-axis.
5. In ref. please use abbreviated journal names.
6. No data availability section. No acknowledgement section.
7. Abstract is too short. It is hard to recognize the novelty and understand proposed method.
8. No future work in Conclusion section. Authors need to summarize important simulated results.
9. Authors need to check authors guidelines for MDPI.
10. In Genertic algorithm, Please correct genertic algorithm.
11. In Genertic algorithm, how to handle that if there is some errors unpredictable ?
12. Analysis in Figure 7 is very limited. What means graph which indicates the fluctuation ?
13. In Figure 8, third Figure, why there is dropped in Eb ?
14. Why authors showed Figure 9 ?
15. In Figure 9, how to handle that if there is unpredictable work flow ?
16. Authors had better show some output work if proposed algorithm is applied as shown in Figure 1A or 1B.
17. Is there any standard deviation graph for Figure 8 ?
18. Future work in Conclusion section is needed.
19. Authors had better compare your measured results with previous studies. If possible, please provide the Table with some references.
20. Please mark the highest points in Figure 8.
21. Please provide city and country information for conference papers.
22. Authors had better show advantages and disadvantages of previous other researchers' work.
Author Response
Dear Reviewer,
The authors of the article is very grateful to you for giving very precious comments so that we can improve the article better. Your comments demonstrate very professional knowledge, we highly appreciate your support. Please see the list of point-to-point responses to your comments in the attached file.
Best regards,
Vu Toan Thang

Reviewer 2 Report
In this paper, the digital twin was used to find the robot's motion trajectory, which helped to prevent human-robot conflicts. The algorithm was designed to reduce the number of moves required to obtain materials and to ensure that the robot always had enough materials to assemble along the defined trajectory, thus saving labor and optimizing the manufacturing process. It has strong theoretical significance and certain innovation, which reflects that the author has a good grasp of the basic theory. But there are still some shortcomings to be further proved. As follows:
1. The font size in Figure 2 is too small.
2. The introduction of the DT method is not clear in Chapter 3.
3. Chapter 3 is too brief.
4. Figure 3 is too simple which needs to be more sophisticated.
5. Figure 9 is not clear enough, lacks of the related figures to illustrate the time spent by workers in movement is reduced.
6. The schematic diagram of the multi-adaptive genetic algorithm doesn't reflect the characteristics of multi-adaptivity.
7. How the digital twin is used in this paper?
8. The optimization of human-robot cooperation is not clear in this paper.
Author Response

(The authors gave the same response as above.)

Round 2
Reviewer 1 Report
Authors improved the manuscript so I recommend the manuscript could be accepted as it is.
Author Response
Dear Reviewer 1,
I am grateful for your valuable feedback and suggestions, which reflect your high level of professionalism, dedication, efficiency, and promptness.
Your comments have greatly contributed to our understanding of our research and enabled us to improve our paper, titled "Applying digital twin and multi-adaptive genetic algorithms in human-robot cooperative assembly optimization” (applsci-2279369).
I would like to express my sincere appreciation for your time and effort in reviewing and providing us with your insightful comments.
Thank you again for your time and consideration.
Sincerely yours,
Vu Toan Thang
Reviewer 2 Report
This paper is a revised paper. I have checked the previous comments, responses and revisions. The authors have revised their paper carefully and tackled most of the points raised by reviewers in the last round of review, based on which I will suggest publication after my new comments are addressed. As follows:
1. The color of blue cells seems like green in Figure 10.
2. Lacks of an response to previous comments.
This paper is a revised paper. I have checked the previous comments, responses and revisions. The authors have revised their paper carefully and tackled most of the points raised by reviewers in the last round of review, based on which I will suggest publication after my new comments are addressed. As follows:
1. The color of blue cells seems like green in Figure 10.
2. Lacks of an response to previous comments.
Author Response
Dear Reviewer,
I am grateful for your valuable feedback and suggestions, which reflect your high level of professionalism, dedication, efficiency, and promptness.
Your comments have greatly contributed to our understanding of our research and enabled us to improve our paper, titled "Applying digital twin and multi-adaptive genetic algorithms in human-robot cooperative assembly optimization” (applsci-2279369).
Below, you will find a list of our point-to-point responses to the reviewers' comments, highlighted in blue.
Thank you once again for your time and valuable feedback.
We eagerly await your response.
Sincerely yours,
Vu Toan Thang
